# Exploring Small-Diameter Melanomas: A Retrospective Study on Clinical and Dermoscopic Features

**DOI:** 10.3390/life13091907

**Published:** 2023-09-13

**Authors:** Maria Fernanda Vianna Hunziker, Beatrice Martinez Zugaib Abdalla, Flavia Vieira Brandão, Luana Pizarro Meneghello, Jaciara Moreira Sodré Hunnicutt, Thais Helena Bello Di Giacomo, Cristina Martinez Zugaib Abdalla, Ana Maria Fagundes Sortino

**Affiliations:** 1Hospital Sirio Libanês, Rua Dona Adma Jafet, 115, Bela Vista, São Paulo 01308-050, SP, Braziljaciara.mshunnicutt@hsl.org.br (J.M.S.H.); thabello@hotmail.com (T.H.B.D.G.); cristina.abdalla@clinicaabdalla.com.br (C.M.Z.A.); ana.m.sortino@dermatis.com.br (A.M.F.S.); 2Dermatology Department, Hospital Regional da Asa Norte, SMHN Quadra 101 Bloco A Área Especial, Brasília 70710-905, DF, Brazil; dirase.central@saude.df.gov.br; 3Universidade Franciscana, Rua Silva Jardim, 1175, Conjunto III, Prédio 17, Sala 809, Santa Maria 97010-491, RS, Brazil; luana.meneghello@ufn.edu.br

**Keywords:** melanoma, small-diameter melanoma, micro-melanoma, mini-melanoma, dermoscopy

## Abstract

**Simple Summary:**

Small-diameter melanomas may escape clinical examination. Both medical and lay individuals do not prioritize small lesions because they usually look for cutaneous melanomas by mole asymmetry, irregular borders, multiple colors, and diameter greater than six millimeters. Clinicians, especially dermatologists, play a pivotal role in improving the early detection of melanoma. The identification of inconspicuous tumors, with a size equal to or smaller than five millimeters, is achievable through comprehensive clinical inspection and dermoscopic examination. By integrating clinical total body exam with handheld dermoscopy of all lesions, regardless of their sizes, and total body skin photography combined with digital dermoscopy and sequential digital dermoscopy imaging of suspicious moles, doctors can improve the early detection of melanoma, thus reducing the risk of diagnostic delays, and alleviating the burden on patients and healthcare systems.

**Abstract:**

Background: Early melanoma detection allows for timely intervention and treatment, significantly improving the chances of favorable outcomes for patients. Small-diameter melanoma (SDM) typically represents an initial growth phase of cutaneous melanoma. One of the challenges in detecting melanoma in their early stage lies in the fact that dermoscopy criteria have been primarily designed for fully developed lesions. Early-stage melanomas may be difficult to detect and possibly even be overlooked or misinterpreted during examinations. Methods: The primary aim of this study was to identify valuable clinical and dermoscopic clues to enhance the detection of SDMs. To achieve this objective, we conducted a comprehensive retrospective analysis, including forty SDMs with a diameter of 5 mm or less. These cases were diagnosed over an 8-year period and were collected from five referral centers across Brazil. Seven experienced dermatologists independently assessed the dermoscopic features of each lesion. Additionally, this study includes demographic and histological information. Results: The study encompassed a total of 28 patients, of which 16 were females, accounting for 58% of the participants, with an average age of 43.6 years. Among the small-diameter melanomas (SDMs) under investigation, the majority, constituting 27 cases (69.2%), were identified as “de novo” lesions, i.e., not associated with a nevus. Additionally, eight SDMs (20%) exhibited invasive characteristics, with Breslow index measurements ranging between 0.2 to 0.4 mm, suggesting an early stage of malignancy. During dermoscopic examinations, the most prevalent features observed were irregular dots and globules, present in 95% and 87.5% of cases, respectively. Moreover, brown structureless areas were identified in 70% of lesions, followed by atypical network (67.5%), pseudopods (55%), dotted vessels (47.5%), flat structureless blue-gray areas (42.5%), and irregular blotches (40%). Notably, all SDM were diagnosed in patients under surveillance through total body skin photography (TBSP) and Digital Dermoscopy (DD). Conclusions: Dermoscopy significantly enhances the diagnostic accuracy of melanoma, even in its early stages. Particularly for high-risk patients with numerous nevi, the identification of a new lesion or subtle changes on dermoscopy during follow-up may serve as the sole clue for an early diagnosis. This emphasizes the critical role of dermoscopy in SDM detection and reinforces the importance of surveillance in high-risk patients for timely and effective management.

## 1. Introduction

Recognizing early forms of melanoma is crucial as it correlates with improved survival rates [1]. When detected early in the disease’s biological course, the adequate excision of the tumor is associated with good prognosis [2]. However, small-diameter melanoma (SDM) is often overlooked on clinical examination because they lack the “D” (diameter) criteria of the ABCDE rule [3]. These lesions typically represent an initial growth phase of cutaneous melanoma, and the paucity of clinical and even dermoscopic features associated with malignancy at this stage makes early diagnosis challenging [4].

SDM constitute a minority of diagnosed lesions, their frequency ranging from 2.4–38.2%, depending on the study methodology and number of lesions included [1,2,5]. The increase in public awareness of melanoma, coupled with the adoption of Total Body Skin Photography (TBSP) and Sequential Digital Dermoscopy Imaging (SDDI) for melanoma surveillance to detect early signs among high-risk individuals [6,7,8], has led to an influx of patients presenting with smaller pigmented lesions, requiring careful differential diagnosis that includes melanoma [8]. Of note, these two photographic approaches, TBSP and digital dermoscopy (also known as dermatoscopy or epiluminescense microscopy) or SDDI, play crucial and complementary roles in the differential diagnosis of malignant melanoma. TBSP offers baseline images to detect new lesions or macroscopic changes of a lesion, while dermoscopy reveals subtle alterations (mostly gain or loss of dermoscopic structures) in preexisting nevi. However, the utilization of both modalities is often limited by time and cost constraints, restricting their use to a selected group of high-risk patients in pigmented skin lesion clinics [9].

While dermoscopic criteria have been extensively described for fully developed lesions, early-stage melanoma can be difficult to detect. Dermoscopic features typically correlated with malignancy, such as atypical pigment network, pseudopods, blue-white veil, multiple colors, irregular dots, and globules, may not be present in SDM [6]. In some cases, changes along clinical and dermoscopic surveillance may be the only feature suggesting malignancy in these small lesions [10]. Dermoscopic features and reproducibility of dermoscopy algorithms for SDM have been sparsely addressed in the existing literature, with only a limited number of publications focusing on the characterization of these lesions [3,4,5,11,12,13,14].

This study aims to clinically and dermoscopically describe histologically confirmed SDM melanoma cases with an in vivo maximum diameter of up to 5 mm. Identifying these features will aid clinicians in making an early diagnosis and improve patient outcomes. Additionally, our findings offer valuable insights that can contribute to the existing body of knowledge, furthering ongoing efforts in research and standardization.

## 2. Materials and Methods

This retrospective, multicenter, cross-sectional, analytical, and descriptive study was conducted with the primary objective of investigating the diverse clinical and dermoscopic characteristics exhibited by 40 SDM. The conceptualization of this study was led by authors CMZA and AMFS, who are coordinators of the postgraduate program in oncodermatology at Hospital Sírio-Libanês (São Paulo, Brazil). The motivation to investigate the dermoscopic characteristics of SDMs stemmed from discussions between professors and students. These discussions were prompted by the challenges faced in identifying these lesions during bedside clinical examinations. Professors and dermatologists involved in the oncodermatology postgraduate program at Hospital Sírio-Libanês provided cases of SDMs for the purpose of examining dermoscopic structures. These cases were gathered from specialized private clinics focused on melanoma screening, utilizing 2D whole body imaging and dermoscopy. These clinics are situated in three states across Brazil, namely São Paulo (3 clinics; *n =* 28, 70% cases), Distrito Federal (1 clinic; *n =* 2, 5% cases), and Rio Grande do Sul (1 clinic; *n =* 10, 25% cases).

In total, the study included 28 patients, and their melanoma diagnoses were made between 2012 and 2020. All SDM cases were identified using a combination of total body skin examination (TBSE), digital dermoscopy (DD), either manual or automated total body skin photography (TBSP), and sequential digital dermoscopic imaging (SDDI).

A collection of digital dermatoscopy images were numbered as cases 1 to 40 and were then randomly compiled and distributed to seven dermatologists for a dichotomous evaluation (dermoscopy experience: CMZA and AMFS with >20 years; MFVH, FVB, LPM with >10 years; JMSH and BMZA with <10 years). This assessment aimed to determine the presence or absence of preselected dermatoscopic structures, as outlined in a table provided to the authors for completion.

The dermoscopic structures assessed for their presence or absence are outlined in Table 1. The selection of dermoscopic features was based on the available literature [15,16,17,18,19], ensuring a comprehensive analysis of the lesions in the study. Given the ongoing debates in the current literature regarding the standardization of dermatoscopic structure terminology, with some authors using descriptive terms and others employing metaphoric terms, we sought to minimize interpretational subjectivity among the authors. This was achieved by providing clear descriptions of dermatoscopic terms and adopting terminology references from the Third Consensus Conference of the International Society of Dermoscopy [19].

There was agreement in the analysis of most criteria among the evaluating dermatologists. In cases of disagreement, the dermoscopic criterion was considered present if 50% plus one dermatologist agreed.

Dermoscopic images were captured using a digital epiluminescence microscope (FotoFinder^®^; Dermoscope Software GmbH, Bad Birnbach, Germany), employing a 20-fold magnification. The instrument and calibration method have been extensively described elsewhere [16]. The camera models used were Medicam 800 and 1000 Full HD.

Demographic data, such as gender and age, along with skin phototypes according to Fitzpatrick’s classification, and anatomical location data were evaluated. Histopathological information including invasion, Breslow thickness, association with preexisting nevus, histopathological subtype, mitosis, ulceration, satellitosis, and perineural- or vascular invasion was also retrieved. Histopathological evaluations were carried out by various experienced dermatopathologists. These professionals are regularly responsible for performing routine histopathological analyses for the private clinics that contributed with the melanoma cases for this study.

The eligibility criteria for inclusion in this study were as follows: lesions with a diameter ≤5 mm, as diameters were measured using FotoFinder^®^ software measurement tool. Additionally, only cases with a confirmed histopathological diagnosis of melanoma were considered. Furthermore, only lesions with available high-quality clinical and dermoscopic images were included. We received a total of 44 cases. Four cases were excluded from the analysis due to low image quality or lack of pathology report.

On the other hand, certain exclusion criteria were applied. Lesions located on the acral site, scalp, nails, and mucosae were excluded from the study. Also, lesions with equivocal histopathological diagnosis, such as collision tumors, SAMPUS (Superficial Atypical Melanocytic Proliferations of Unknown Significance), MELTUMP (Melanocytic Tumors of Uncertain Melanocytic Potential) [17], Atypical Spitz Tumor were excluded from the analysis.

The study adhered to the principles outlined in the Declaration of Helsinki [18], ensuring ethical guidelines and patient welfare. To maintain confidentiality, all the images are anonymous and not identifiable. Prior to image capture, patients provided written informed consent be photographed within their respective outpatient dermatologic clinic.

We studied the clinical-demographic findings and dermoscopic characteristics based on tumor thickness (in situ and invasive melanomas). The information from quantitative variables was summarized using mean, standard deviation, median, minimum, and maximum values, as well as the count of valid observations. Information from qualitative variables was summarized through simple frequency and percentage.

For the statistical analysis, IBM SPSS Statistics 22 for Windows was selected as the tool. To analyze clinical-demographic findings and dermoscopic characteristics based on tumor thickness (in-situ and invasive melanomas), the nonparametric Mann–Whitney test was used for quantitative variables, and the Chi-squared test, or if necessary, the Fisher’s Exact test or the Likelihood Ratio test, was used for qualitative variables. A significance level of *p* < 0.05 was applied to determine statistical significance.

## 3. Results

The study involved forty SDMs in a cohort of twenty-eight patients (*n =* 28), consisting of sixteen females and twelve males, with ages ranging from 32 to 79 years old (mean 46.3). Fitzpatrick’s skin Phototype I showed a higher percentage of SDM cases in our cohort. Regarding lesion locations, the back and legs had the highest number of SDMs. Detailed characteristics of the cases with clinical-demographic findings are presented in Table 2.

Among the SDMs with a diameter ≤ 5 mm (ranging from 1 to 5 mm) the majority (27; 69.2%) were not associated with preexisting nevi, indicating that they were “de novo” lesions. Out of the total cases, thirty-two (80%) were identified as melanoma in-situ, while the remaining eight cases (20%) were invasive, with a Breslow index not exceeding 0.4 mm, suggesting an early stage of malignancy.

Unfortunately, the study faced limitations as histological subtype information was not provided in a representative number of histopathological reports (19; 47.5%). Among the cases with available subtype information, the superficial spreading type was the most prevalent, comprising eighteen cases (45%). Regarding the invasive melanomas, none had mitosis, ulceration, regression, or vascular or perineural invasion. Table 3 presents histopathological findings.

Out of the total lesions evaluated, 90% displayed pigmentation, while 10% showed hypo/amelanotic characteristics. Some representative dermoscopic images of the included SDMs are depicted in Figure 1 and Figure 2.

A comprehensive evaluation of dermoscopic features, including melanoma-specific characteristics, was conducted. Notably, the most prevalent dermoscopic features were irregular dots and globules (95 and 87.5%, respectively). Structureless brown areas were also commonly observed, accounting for 70% of cases. Additionally, atypical network was detected in 67.5% of lesions. Atypical streaks/pseudopods were evident in 55% of cases, while 42.5% displayed distinctive flat structureless blue-white areas. Irregular blotches, as off-centered hyperpigmented areas, were observed in 40% of cases. Regression/scar-like depigmentation, granularity (peppering), angulated lines, and shiny white structures were dermoscopic features less frequently encountered. Moreover, dotted vessels were identified in 47.5% of cases, with 30% displaying polymorphous vessels. For a more comprehensive visualization of the distribution of dermoscopic features, Figure 3 visually represents their prevalence.

Table 4 illustrates the level of consensus among the dermatologists who evaluated the images, based on their alignment to one dermatologist with over 20 years of experience, regarded as benchmark. According to the results in Table 4, at a significance level of 5%, there was variable levels of agreement between dermatologists only for 13 specific dermoscopic structures.

## 4. Discussion

The diagnostic challenge posed by SDMs is significant since clinicians often overlook them during examinations, primarily focusing on lesion size for diagnosis and dermoscopy assessment [4]. Many clinicians take size into consideration for clinical judgement of a pigmented lesion, and lesion diameter is a factor for preselecting lesions for dermoscopy evaluation. Seidenari et al.’s study [4] of 482 melanomas revealed that 16% had a diameter ≤6 mm, and 5% measured <4 mm. Surprisingly, half of the small melanomas exhibited clinical symmetry and were not visually attention-grabbing from a clinical perspective. This accents the need for enhanced vigilance in detecting and evaluating small melanocytic lesions.

Irrespective of the frequency of SDM occurrence, which can vary based on study methodology and whether size measurements were performed clinically on pathological specimens [1,2,5,10], there appears to be a trend towards diagnosing smaller lesions. This shift is likely attributed to routine adoption of dermoscopy and the widespread use of TBSP with SDDI. A recent retrospective study comparing melanomas diagnosed at a dermatology unit in Milan, Italy, from 2006 to 2020, found a significant increase of in-situ melanomas and small-diameter melanomas (<6 mm) over the studied period. Specifically, during the first biennium, 27 out of 220 (12.3%) cases were identified as SDM, while in the last biennium, 61 out of 236 (25.9%) cases were classified as SDM. This highlights the growing capacity of SDM detection over time [19].

In our study, comprising 40 cases of melanomas with a diameter ≤5 mm, we observed a notable shift in diagnosis patterns over time. Between 2012 and 2016, only 6 cases (15%) were diagnosed, while between 2017 and 2020, a substantial increase to 34 cases (86%) was evident. This observed surge in diagnoses can likely be attributed to the heightened attention directed towards small melanocytic lesions since 2017. The growing emphasis on early detection and awareness of the clinical significance of smaller lesions has likely led to more frequent biopsies, subsequently contributing to the increased detection rate [19].

The increasing trend of encountering smaller atypical melanocytic lesions presents significant histological diagnostic challenges for dermatopathologists. Consequently, there is a possibility that certain early melanomas might be inadvertently diagnosed as benign, and vice-versa [20]. Caution is important when interpreting architectural disorder in small lesions to avoid overdiagnosis of melanoma [20]. Especially in cases with a diameter of ≤4 mm, diagnostic disagreements among pathologists are not unlikely [15]. To address this complex issue, Ferrara et al. [21] propose a collaborative “consensus diagnosis” approach among pathologists, emphasizing the necessity of pooling collective expertise to resolve diagnostic uncertainties. They also highlight the importance of intense atypia cytology as a valuable indicator for accurately diagnosing melanomas with a diameter ≤4 mm, providing guidance for identifying these difficult cases.

In light of the above information, when dealing with small melanocytic tumors showing suspicious clinical and dermoscopic features, challenges emerge for clinicians and pathologists alike. Several studies have documented a correlation between increased melanoma incidence and higher biopsy rates [22]. The phenomenon of increased diagnoses without a proportional rise in mortality may be attributed to earlier detection and improved treatment. However, it is essential to recognize that increased screening can also lead to more melanoma diagnosis. While some lesions contributing to melanoma incidence are considered biologically benign due to false-positive results in the early stage, there was indeed an upward trend in melanoma cases and thicker lesions [22,23].

It is important to emphasize that scenarios involving high-risk patients [24] with new lesions (“de novo”), lesions displaying clinical or dermoscopic changes during follow-up, or those with high index of clinical suspicion, combined with severe atypia on the histological section, but lacking clear-cut melanoma histology, require careful management. In such instances, the authors’ approach is to treat these lesions as melanomas, considering the potential implications for the patient’s health and prognosis. This cautious approach ensures that suspicious or potentially malignant lesions are not overlooked, prioritizing patient safety and early intervention when warranted.

In our study, we observed a range of prevalent dermoscopic structures among the cases of SDMs. The most frequently encountered dermoscopic features were atypical dots (95%) and atypical globules (87.5%). Additionally, structureless brown areas were present in 70% of cases, followed closely by the atypical pigment network in 67.5% of cases. Furthermore, atypical streaks/pseudopods were identified in 55% of cases, while 42.5% exhibited flat blue-white structureless areas. Irregular hyperpigmented areas (atypical blotches) were observed in 40% of cases.

The intricate diagnosis of small-diameter atypical pigmented lesions based on dermoscopic examination stress the importance of a comprehensive approach, incorporating clinical evaluation, selective utilization of confocal microscopy [25], and histopathological examination. Pupelli et al.’s study [26] underscores the significance of confocal microscopy in distinguishing small melanomas from naevi, with melanomas displaying cytological atypia and irregular nesting, and naevi showing a more regular architecture and occasional mild cytological atypia. Further studies focusing on the analysis of suspicious small-diameter pigmented lesions using confocal microscopy are necessary. These studies can help establish a strong correlation between the histopathological findings and the dermoscopic features observed in these lesions. Such investigations will provide valuable insights into refining diagnostic criteria and enhancing the accuracy of differentiation between early-stage melanomas and naevi.

In this study, we identified dotted vessels in 47.5% and polymorphous vessels in 30% of the cases. These dermoscopic features play a significant role in the evaluation of SDMs. Interestingly, Nazzaro et al. [14] observed atypical vessels exclusively in invasive melanomas with a diameter of ≤5 mm, although the association did not achieve statistical significance. Similarly, Regio Pereira et al. [10] reported an association between atypical vascular patterns, macular blue-whitish areas, and shiny white structures with invasive small-diameter melanomas.

It is worth noting that shiny white structures, regression structures, e.g., white structureless areas (scar-like depigmentation), and gray, blue, or black dots (peppering) were infrequently encountered in our study. The structure blue-white veil was not seen in the cases studied. These findings are consistent with previous publications as these dermoscopic structures are typically associated with more advanced melanomas [4].

The literature on SDMs exhibits significant heterogeneity, making comparisons between studies challenging. Different methodologies, variations in size measurements (particularly when performed on pathological specimens), and the absence of a universally accepted definition for what constitutes a small diameter contribute to the complexity of this research area.

For instance, Megaris et al. [3] reported on global dermoscopic features in melanomas with up to 5 mm diameter, observing frequencies of reticular (57.7%), structureless (26.9%), globular and reticular (11.5%), and spitzoid (3.8%) patterns. In another study by Seidenari et al. [4], six global dermoscopic patterns were identified in a group of 22 micro-melanomas (≤4 mm diameter), including multicomponent (32%), bicomponent (27%), spitzoid (18%), reticular (14%), and globular (5%). The Spitzoid pattern emerged as the most frequent in another study [10]. Remarkably, a structureless global dermoscopic pattern was noted in a significant majority of cases, constituting 65% of the study cohort.

In a Brazilian study [13], out of 481 suspicious lesions, 123 were micro-melanomas, and the most prevalent dermoscopic structures exhibited brown color (84.5%), structureless areas (86.2%), atypical pigmented network (62.6%), and asymmetric dots (61%). De Giorgi et al. [27] also conducted a study on small melanocytic lesions under 6 mm in diameter, revealing a higher occurrence of atypical networks, atypical globules, and streaks in melanomas compared to naevi. However, the atypical vascular pattern was less prevalent in melanomas, representing only 12% of cases (four cases). In another study involving 96 small, equivocal lesions, researchers used the seven-point checklist and found a significant association between the atypical vascular pattern and melanoma diagnosis [26]. The variation in dermoscopic devices (polarized vs. nonpolarized dermatoscope) and image acquisition pressure could explain the apparent discrepancy in the study’s results. As mentioned previously, vascular structures were observed in a significant number of cases in our study. This observation might be attributed to the higher proportion of fair-skinned individuals among the patients. However, we were unable to establish a correlation between vascular structures and invasive lesions.

In epidemiologic studies, it is generally observed that patients with cutaneous melanoma fall within the age range of 55 to 65 years old [28,29]. However, our study’s sample exhibited a relatively lower average age, with patients having an average age of 46.3 years old.

Moreover, despite the small diameter of the tumors in our study, a significant proportion (20%) exhibited invasive characteristics. These findings align with other research, which has also demonstrated that a small diameter does not exclude the possibility of invasion. Notably, small-diameter invasive melanomas constitute a substantial portion, accounting for approximately 30% of all invasive melanomas [10]. Megaris et al. [3] reported 27% of invasive lesions among melanomas with a diameter ≤5 mm, and in another study of 206 melanocytic lesions with a diameter ≤3 mm, 23 were identified as melanomas, with 19 being invasive.

A recent Greek publication [30] shed further light on SDM, revealing that out of 537 invasive melanomas, 57 (10.6%) were classified as SDMs (<6 mm in greatest diameter on histological section). Interestingly, they found no significant difference in the median Breslow thickness between small and larger diameter melanomas (0.8 mm vs. 0.9 mm, respectively). Additionally, of the SDM group, five (8.9%) presented with metastasis. These findings emphasize the importance of recognizing smaller melanomas as they may exhibit invasive and aggressive behavior.

In a recent multicenter retrospective study [14], comprising 269 cases, including 103 SDMs (≤5 mm, flat, non-facial), 81 common melanocytic nevi (≤5 mm), and 85 control large (>5 mm) melanomas, demographics, clinical, and dermoscopic features were evaluated. The study found that SDMs were associated with a younger age at diagnosis and a preference for upper limbs. Dermoscopic features like atypical pigment network, blue-white veil, pseudopods, peripheral radial streaks, and the presence of multiple colors were identified as melanoma predictors for small lesions (≤5 mm). Among the SDM group, 44 (42.7%) lesions were in-situ, and blue-white veil and negative pigment network were associated with invasion.

Finally, in Drugge et al.’s study [9], 28 micro-melanomas were analyzed, and all lesions were diagnosed using a routine comparison of complete sets of Total Body Photography (TBP) images combined with dermoscopy. Some of the lesions were already invasive. The study described that dermoscopic features such as chaos, clods, and amorphous areas were identified in all malignant lesions.

As more studies explore SDM features, the diversity in reported dermoscopic patterns described in different publications presents an opportunity to better understand and improve diagnosis of these lesions.

The current study has some limitations that should be taken into consideration. Firstly, the sample size is relatively small, which may affect the generalizability of the findings to a broader population. Additionally, the study design being retrospective in nature might introduce potential biases and limitations in data collection and analysis.

Another potential limitation is the lack of control groups for comparison, which could have provided more robust insights into the observed findings. The inclusion of high-risk patients from specialized dermatologic clinics may have led to a selection bias, potentially impacting the representation of the broader population.

Furthermore, as discussed above, differentiating between SDM and atypical nevi histologically can be challenging, and misclassification of some tumors is possible. As this is a retrospective study, the cases were assessed by different pathologists, resulting in a lack of uniformity in the histopathological evaluation. However, all of them are experienced professionals in dermatopathology, typically referred to for cases from the involved clinics. It is important to note that all of these pathologists hold board certification in the field. To avoid the inclusion of misclassified lesions, we excluded doubtful histopathological cases.

While our study may be modest in scale, it provides valuable insights for future research undertakings. When combined with previous publications that have outlined the dermoscopic features of such small melanomas, these collective findings markedly enhance our understanding and improve the identification of SDMs. Exploring these aspects facilitates a more comprehensive understanding of the diagnostic challenges posed by small melanocytic tumors.

Clinicians play a pivotal role in improving the early detection of melanoma, thus reducing the risk of diagnostic delays and alleviating the burden on patients and healthcare systems. The identification of SDMs with a size of ≤5 mm is achievable through comprehensive clinical inspection and dermoscopic examination. SDMs can often be inconspicuous during clinical examination, and without the aid of dermoscopy, they may go unnoticed. Unfortunately, many dermatologists do not prioritize small lesions and may neglect to conduct dermoscopic evaluations.

By integrating dermoscopy into routine examinations, clinicians can significantly improve their diagnostic accuracy, leading to earlier detection and prompt intervention when necessary. Suboptimal diagnostic accuracy of SDMs may result from the incomplete development of specific melanoma features on dermoscopy [10]. Close follow-up with clinical photographs and dermoscopy in high-risk patients with numerous nevi is crucial [24,31]. Early detection relies on identifying new and changing lesions, with the assistance of TBSP and SDDI [10]. Subtle changes observed during follow-up examinations may serve as the only indicative sign of an early-stage melanoma [32].

In conclusion, we emphasize the importance of utilizing handheld dermoscopy to examine all suspicious lesions, irrespective of their size. Furthermore, we underscore the significance of continuous monitoring of high-risk patients using TBSP, DD, and SDDI. Lastly, despite the limitations of our study, the findings contribute to the comprehension of SDMs and their identification. When a small-diameter “de novo” melanocytic lesion shows one or more of the following melanoma specific structures: atypical dots and globules, brown structureless areas, atypical pigment network, atypical streaks/pseudopods, blue-white structureless areas, atypical blotches, and vessels as dotted, linear or polymorphous, an SDM has to be included in the differential diagnosis and a biopsy is advised. These insights lay valuable groundwork for future research in this domain, highlighting the need for additional studies to corroborate and build upon these observations.

## Figures and Tables

**Figure 1 life-13-01907-f001:**
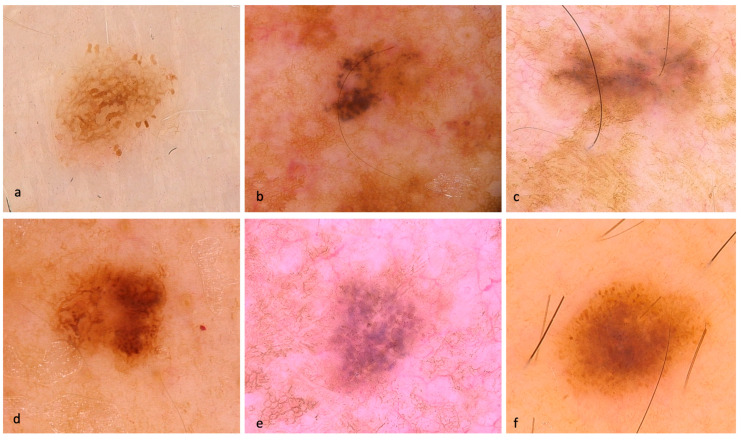
Dermoscopic images of SDMs. (**a**) in-situ SDM with atypical pigment network, atypical globules, and pseudopods. (**b**) in-situ SDM with atypical globules, dark blotches, and a small central milky red area. (**c**) in-situ SDM with structureless pattern, milky red areas, and granularity/peppering. (**d**) in-situ SDM with atypical pigment network, focal atypical globules, and structureless central brown area. (**e**) invasive SDM (Breslow thickness of 0.3 mm) with patchy peripheral atypical pigment network, blue-gray atypical globules, granularity, negative network, and targetoid dots. (**f**) invasive SDM (Breslow thickness of 0.4 mm) with teared globules around the entire lesion, focal pseudopods, and central structureless brown areas.

**Figure 2 life-13-01907-f002:**
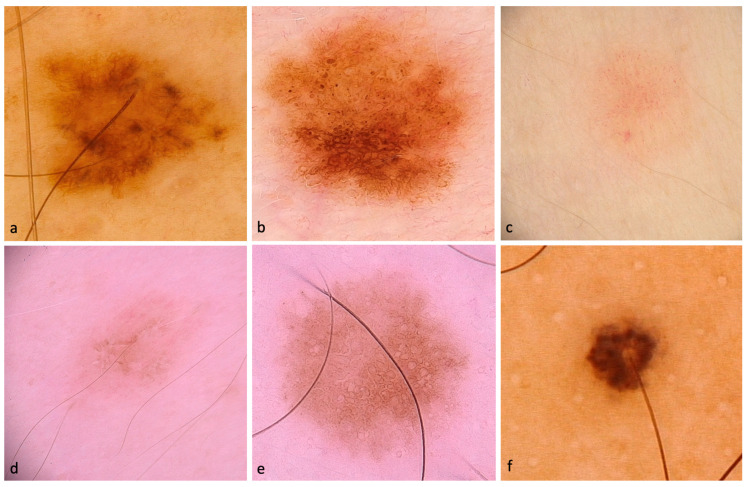
Dermoscopic images of SDMs. (**a**) in-situ SDM with patchy peripheral atypical pigment network, tan structureless areas, and multiple small hyperpigmented areas. (**b**) in-situ SDM with atypical pigment network and atypical globules (**c**) invasive SDM (Breslow thickness of 0.2 mm) with polymorphous vessels. (**d**) in-situ SDM with negative network, atypical globules, targetoid dots, and brown circles. (**e**) in-situ SDM with atypical network and brown circles. (**f**) in-situ SDM with atypical globules and hyperpigmented blotches around the hair follicle.

**Figure 3 life-13-01907-f003:**
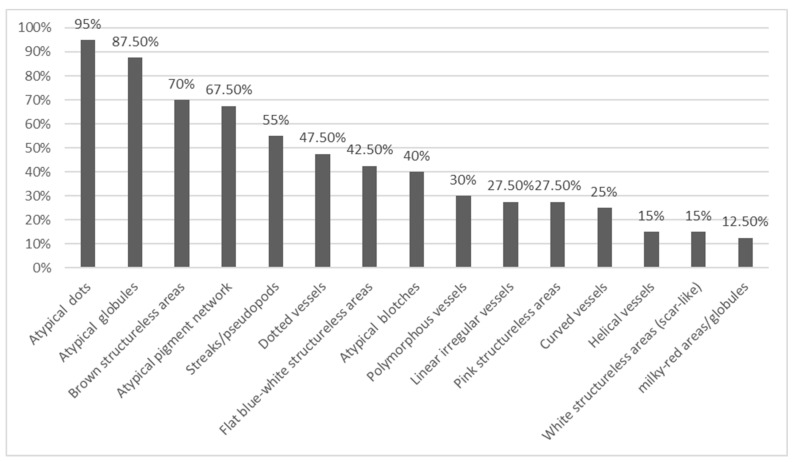
Shows the frequency of 15 most prevalent dermoscopic structures of forty SDMs studied.

**Table 1 life-13-01907-t001:** Evaluated Dermoscopic Characteristics by Metaphoric and Descriptive Terminology.

Dermoscopic Metaphoric Terminology	Dermoscopic Descriptive Terminology
1. Atypical pigment network	Lines, reticular and thick or reticular lines that vary in color, irregular
2. Patchy peripheral atypical pigment network	Peripheral or focally distributed lines, reticular and thick, or varying in color
3. Angulated lines/Polygons	Lines, angulated or polygonal (non-facial skin)
4. Negative network	Lines, reticular, hypopigmented, around brown clods
5. Atypical Streaks/Pseudopods	Lines, radial (always at periphery), irregular/Bulbous projections at the lesion edge, either directly associated with a network or tumor border
6. Atypical dots	Any distribution of dots other than at the center of the lesion, or located on the network lines, irregular
7. Atypical globules	Clods with variability in color, size, shape, or spacing and distributed in an asymmetric fashion, irregular
8. Tiered globules	Clods distributed at the periphery of lesion, in layers
9. Tan structureless areas	Structureless areas, light brown (tan), eccentric
10. Prominent skin markings	Crossing Linear Furrows, lighter in color than the rest of the lesion
11. Blotch/Small blotches	Structureless zone, brown or black/Multiple small hyperpigmented areas
12. Atypical blotch	More than one blotch or a blotch that is located off-center, irregular
13. Flat blue-white areas	Flat structureless zone, blue-white
14. Blue-white veil	Raised structureless zone, blue-white
15. Granularity/peppering	Dots, gray, blue or black
16. Regression/scar-like depigmentation	Structureless zone, white
17. Shiny white blotches and strands/streaks *	Clods, white, shiny/Short, discrete white lines arranged both parallel and perpendicular to each other *
18. Targetoid dots	Dots, brown, central (in the center of hypopigmented spaces between reticular lines)
19. Brown circles	Hyperpigmented microcircles, focal and off- centered
20. Homogeneous brown	Structureless brown areas
21. Homogeneous pink	Structureless pink areas
22. Milky-red areas/globules	Clods, pink or pinkish-white, and small
23. Comma vessels	Vessels, curved
24. Dotted vessels	Vessels, dots
25. Serpentine vessels	Vessels, linear
26. Corkscrew	Vessels, helical
27. Polymorphous vessels	Vessels, polymorphous

* Only visible by polarized dermoscopy.

**Table 2 life-13-01907-t002:** Clinical-demographic findings according tumor thickness (in-situ and invasive SDM).

Small-Diameter Melanoma	In Situ	Invasive	Total	*p*-Value
Sex				
Female	18 (56.3%)	6 (75%)	24 (60%)	0.439 *
Male	14 (43.8%)	2 (25%)	16 (40%)	
Age				
Average (Standard deviation)	47.3 (13.2)	41.6 (11.2)	46.3 (12.9)	0.257 **
Median age (25 percentile–75 percentile)	42 (37–54)	34 (33–54)	41.5 (36–54)	
Fitzpatrick skin phototype				
I	7 (21.9%)	7 (87.5%)	14 (35%)	0.001 ***
II	15 (46.9%)	1 (12.5%)	16 (40%)	
III	10 (31.3%)	0 (0%)	10 (25%)	
Location				
Abdomen	1 (3.1%)	1 (12.5%)	2 (5%)	
Back	9 (28.1%)	1 (12.5%)	10 (25%)	0.846 ***
Trunk	5 (15.6%)	1 (12.5%)	6 (15%)	
Forearm	3 (9.4%)	1 (12.5%)	4 (10%)	
Leg	6 (18.8%)	1 (12.5%)	7 (17.5%)	
Arm	5 (15.6%)	1 (12.5%)	6 (15%)	
Thigh	3 (9.4%)	2 (25%)	5 (12.5%)	
Lesion Size (mm)				
Average (Standard deviation)	3.1 (1.3)	4.1 (0.6)	3.3 (1.3)	0.119 **
Median (25 percentile–75 percentile)	3.5 (2–4)	4 (3.5–4.5)	3.5 (3–4)	

* Exact Fisher test; ** Mann–Whitney; *** Likelihood ratio.

**Table 3 life-13-01907-t003:** Histologic findings.

Small-Diameter Melanoma	In Situ	Invasive	Total	*p*-Value
Histologic subtype				
Lentiginous melanoma	3 (9.4%)	0 (0%)	3 (7.5%)	<0.001 ***
Non-specific	19 (59.4%)	0 (0%)	19 (47.5%)	
Superficial spreading melanoma	10 (31.3%)	8 (100%)	18 (45%)	
Total	32 (100%)	8 (100%)	40 (100%)	
Nevi-associated				
No information	1	0		
Compound	5 (16.1%)	1 (12.5%)	6 (15.4%)	0.857 ***
Intradermal	2 (6.5%)	1 (12.5%)	3 (7.7%)	
Junctional	2 (6.5%)	0 (0%)	2 (5.1)	
“de novo” melanoma	21 (67.7%)	6 (75%)	27 (69.2%)	
other	1 (3.2%)	0 (0%)	1 (2.6%)	
Total	31 (100%)	8 (100%)	39 (100%)	
Breslow thickness				
0.2	0 (0%)	3 (37.5%)	3 (7.5%)	not calculated
0.3	0 (0%)	2 (25%)	2 (5%)	
0.4	0 (0%)	3 (37.5%)	3 (7.5%)	
N/A	32 (100%)	0 (0%)	32 (80%)	
Total	32 (100%)	8 (100%)	40 (100%)	

*** Likelihood ratio.

**Table 4 life-13-01907-t004:** Agreement of 13 Dermoscopic Structures with Benchmark Dermatologist.

Dermoscopic Findings by Dermatologists ^1^	Agreement Level ^#^				
	0–0.20 (Minimum)	0.21–0.40 (Reasonable)	0.41–0.60 (Moderate)	0.61–0.80 (Substantial)	0.81–1.0 (Perfect)
Atypical dots		****			
Atypical globules		*	***	**	
Structureless areas, brown		*	**		
Atypical pigment network	*	*	****		
Atypical Streaks/Pseudopods		**	**		
Dotted vessels		*	**	*	
Flat structureless areas, blue-white			**		
Atypical blotches		**	***	*	
Polymorphous vessels				*****	*
Vessels, linear and irregular				****	
Structureless areas, pink			****	**	
Vessels, curved		**			
Vessels, helical				*	

^1^ Seven dermatologists have examined dermoscopy images. ^#^ The Kappa coefficient value was determined, and one dermatologist with over 20 years of experience was chosen as benchmark for comparisons. The significance level adopted is 5% (*p*-value ≤ 0.05). * A single dermatologist agreed with benchmark. ** Two dermatologists agreed with benchmark. *** Three dermatologists agreed with benchmark. **** Four dermatologists agreed with benchmark. ***** Five dermatologists agreed with benchmark.

## Data Availability

The data from the study is included within the article itself.

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
