# Peer review of "Exploring Small-Diameter Melanomas: A Retrospective Study on Clinical and Dermoscopic Features"

_life, 2023, doi:10.3390/life13091907_

Round 1

Reviewer 1 Report

In a well-written manuscript, the authors make a valuable contribution to a field where much knowledge is still lacking.

An increasing number of the world's dermoscopists do not use metaphorical terms but instead, and for many almost exclusively, use revised pattern analysis and descriptive language to describe patterns and specific clues in the diagnosis of melanoma. The results and interpretation of the study should be made available and understandable also for them to evaluate.

When the dermoscopists in the study have reached a consensus on the presence of a specific structure, the authors should describe how it was previously defined for the group and how this definition was communicated among them. Terms such as irregular and atypical are particularly prone to subjectivity.

The terms atypical and irregular are sometimes used interchangeably in the text, how are these terms defined and what is the difference between them?

In some places the text uses both metaphorical and descriptive language to name structures and patterns in a complementary way. This is good but should be done systematically throughout the text, or at least the metaphorical terms should be clearly defined and preferably translated at some point in the text.

Although the terms may be defined in the references, the definitions are important and crucial for the interpretation of the work and should be specifically defined in this manuscript.

Author Response

I sincerely appreciate your review and the thoughtful suggestions you provided for the refinement of the work. Your insights have been taken into careful consideration. In response, we have provided a more detailed elaboration of dermatoscopic structures used for the dichotomous analysis of the evaluated dermatoscopic images within the Methods section, offering a more comprehensive description of these structures. This effort is pivotal in mitigating potential ambiguity or subjectivity in the interpretation of these structures by the participating dermatologists, as well as by the readers of the study. Moreover, recognizing the ongoing debate surrounding the metaphoric and analytical terminologies within the realm of dermatoscopic structural nomenclature, we have made an effort to address this matter within the text itself.

Once again, I extend my gratitude for your valuable feedback.

Reviewer 2 Report

More and more attention is paid to micromelanomas, hence the publication has a non-technical and practical significance and should be published.

Author Response

I appreciate your review. Please find attached the revised manuscript.

Reviewer 3 Report

This is a topic of increasing interest as many studies showed that incidence of SDM is higher nowadays, thanks to pur diagnostic ability (dermoscopy firstly). Also the patients are sensitive with skin cancers and more precise in get controlled by dermatologists and indicating new skin lesions or any changes. This in fact could be the reason of high incidence of SDM in upper limbs, as some studies have shown.

Please note that micromelanoma indicate lesions under 3 mm (minimelanoma for lesion smaller than 5 mm).

Please note that "imagens" is not English (I think Portuguese). Add the staging and Breslow of pictures 1 and 2.

Please note that if stagingbis in situ and pt1a ulceration and satellitosis should be always zero.

The discussion is good. The main problem with this paper os the lack of a control group. It could be interesting to know if your case serie can validate the recently published "five point checklist' by Nazzaro.

Author Response

We sincerely appreciate your meticulous review of our manuscript and the valuable insights you provided for its revision. Your feedback was important to refine both the study design and the presentation of methods to ensure greater clarity. We have taken your feedback seriously and have made efforts to improve the Methods section, aiming to enhance the comprehensibility of the applied methods within the study.

Indeed, we acknowledge the inherent limitations within our work and have tried to address these limitations in the Discussion section. Despite the relatively modest scope of the study, our findings emphasize the critical importance of investigating the dermatoscopic aspects of small-diameter melanomas. The challenges presented by small atypical melanocytic lesions in clinical dermatology, owing to the introduction of dermatoscopy into routine diagnostics cannot be underestimated and further studies are needed. Similar to other publications cited in the Discussion section, the presentation of findings regarding the dermoscopic features of these diminutive lesions holds the potential to significantly improve their characterization.

We extend our gratitude for your evaluation. Your commitment of time and dedication to reviewing our work is greatly appreciated.

Please find attached the revised manuscript.

Warm regards,

Maria Hunziker and co-authors
